# Evaluation of Effective Energy Values of Six Protein Ingredients Fed to Beagles and Predictive Energy Equations for Protein Feedstuff

**DOI:** 10.3390/ani14111599

**Published:** 2024-05-29

**Authors:** Qiaoru Zhang, Haoran Sun, Zuer Gao, Hui Zhao, Zhangrong Peng, Tietao Zhang

**Affiliations:** Institute of Special Animal and Plant Sciences, Chinese Academy of Agriculture Sciences, Changchun 130112, China; sarahzhang96@163.com (Q.Z.); solomoncat@163.com (H.S.); gze1010@163.com (Z.G.); baobeihuihui815@163.com (H.Z.); pzr13541494656@163.com (Z.P.)

**Keywords:** pet food, protein, dogs, predictive equations, effective energy values, metabolizable energy, net energy

## Abstract

**Simple Summary:**

Simple Summary: Protein ingredients play a significant role in pet food and have been brought into focus by pet owners, which drives up the annual cost of protein sources for the pet food industry. Protein resources are a crucial part of human food. However, there is a global shortage of protein-rich food, particularly animal protein. Researching the nutritional value and metabolic properties of various protein sources is essential for the pet food industry to make environmentally sustainable pet food. Using the beagles as the model animal and the difference method, we evaluated the energy value of six protein ingredients, including fish meal, meat and bone meal, corn gluten meal, soybean meal, mealworm meal, and yeast extract, and compared the effects on biological potency. There is a correlation between the chemical composition and the effective energy of protein ingredients. This study also provided recommended predictive energy equations for protein ingredients in dog food.

**Abstract:**

This study evaluated the nutrition composition, the nutrient digestibility, and the energy value of six protein ingredients used in pet food by the difference method in six beagles within a 7 × 6 incomplete Latin square design. The results showed that the apparent total tract digestibility of gross energy (GE) and organic matter (OM) in beagles fed the fish meal (FM) and corn gluten meal (CGM) diets was higher than for those fed the meat and bone meal (MBM), soybean meal (SBM), mealworm meal (MM), and yeast extract (YE) diets (*p* < 0.05). The digestible energy (DE), metabolizable energy (ME), and net energy (NE) of the MM diet were greater than the other diets, and MBM was the lowest (*p* < 0.05). The ME of protein ingredients was positively correlated with organic matter and negatively correlated with the ash content. The NE of protein ingredients was positively correlated with the crude protein content and negatively correlated with the ash content. The study resulted in predictive energy equations for protein ingredients that were more accurate than the NRC’s predictive equation of ME when the ash content of the ingredient was more than 30% DM. In conclusion, the nutrient digestibility and energy value of corn gluten meal were similar to those of fish meal and those of soybean meal were similar to yeast extract. All predictive energy equations for six protein feedstuffs had slight differences with measured energy values.

## 1. Introduction

Pet food production was 34.96 million tons all over the world in 2023, and it was still increasing while most of other animal feed was decreased [1]. Companion animals are always provided the best by their owners, whose demand for high-quality pet food is very high [2]. One key aspect that pet food owners and manufacturers consider is the protein source and content [3]. The content of protein in pet food is high, which in canine food ranges from 17.3% to 36.6% [4]. To satisfy the standard of pet food as human food, some companies and some owners choose not to use or feed products containing by-product meal [5]. Some pet foods do not contain plant protein, and some pet owners have expressed concern that gluten in grains may be a source of allergies in dogs [6]. So, it is a big cost of animal protein originally supplied to humans in pet food. Attention to the environment, animal welfare, and climate change are encouraging institutions and individuals to seek alternatives to conventional animal proteins.

Along with the 32,000-year history of the parallel evolution between dogs and humans and adapting to agricultural-based living conditions, dogs evolved from carnivores to omnivores due to large changes in their food source [7]. Both animal ingredients and plant ingredients containing large amounts of protein and starch can be digested and absorbed by dogs [8,9]. To save animal protein-sourced protein and protect the environment, it would be a better way to use more plant-sourced protein and recycled protein to replace part of animal-sourced protein in pet food.

Traditional animal protein, such as chicken, fish, and meat by-products, and plant proteins, such as soybean meal and corn gluten meal, have been the major protein ingredients used in the formulation of commercial pet foods [10,11,12]. In recent years, more novel sustainable ingredients with high protein, such as insect meal and single-cell protein, have also entered the pet food market [11,13]. Yeast extract is the water-soluble extract produced from yeast waste streams, such as *Saccharomyces cerevisiae*, and separated from inner yeast cells. It could be a functional source of nutrients, since yeast extract is rich in proteins, amino acids, nucleotides, sugars, and a variety of trace elements [14]. Lin et al. showed that yeast products may be beneficial to adult dogs by positively altering the gut microbiota, enhancing immune capacity, and reducing inflammation [15]. Because of insect proteins’ low land use, lower greenhouse gas emissions, and low water pollution, they may contribute to sustainable food production as an alternative source of animal protein. Mealworm meal is a kind of by-product after yellow mealworms (*Tenebrio molitor*) larvae defatted, which has a high quality and quantity of protein and amino acids [16,17]. Pet food with insect-based ingredients was poorly accepted for human consumption to feed their pets. Insect-based pet foods proved to be attractive for purchase only when consumers were well informed about the product’s properties in terms of sustainability and healthiness for their pets [18]. So, it is necessary to analyze different protein ingredients in pet food comprehensively and systematically to help pet food companies and pet owners know the utilization of these environment-friendly protein sources.

Knowledge of the energy values and digestibility of ingredients is important to correctly balance pet food [19,20]. Current research on the effective energy value of pet food typically recommends the modified Atwater equation or predictive equations based primarily on fixed energy values and digestibility coefficients associated with the chemical composition of diets to estimate the metabolizable energy (ME) content of pet foods [20,21,22,23]. However, the equations do not apply to all ingredients and may overestimate the food energy of animals [24,25]. The effective energy value of pet food is based on the ME energy system. In opposition to ME, net energy (NE) is a more precise evaluation of the true energy value of the feed [26], because it takes the heat increment (HI) from the digestion and metabolism of feeds into account [27,28]. We can find the rule of energy metabolism of different protein ingredients by total heat production (THP) and HI, which would be useful for losing weight in pets and patients during nutrition recovery.

The most accurate method to assess the effective energy value of feed is to evaluate the animal’s real digestive and metabolic conditions in vivo. The difference method is more suitable for determining the nutrient digestibility and the effective energy value of single ingredients in vivo [29,30,31,32]. A 30% substitution ratio in the difference method in vivo has been shown to effectively assess the energy content of poultry by-product meal for beagles in our previous study [20]. Traditional protein sources in pet foods include poultry by-product meal, fish meal (FM), meat and bone meal (MBM), corn gluten meal (CGM), and soybean meal (SBM) [33,34,35]. Recently, high-quality sustainable protein resources such as insect meals and yeast products have also been used in pet foods [36,37,38].

This study aimed to determine the effective energy values of FM, MBM, CGM, SBM, mealworm meal (MM), and yeast extract (YE) by using the difference method, measure the nitrogen metabolism and heat production, and assess the feces score for beagles. Through stepwise regression analysis of the measured energy value and chemical composition of ingredients fed to beagles, we also derived predictive equations for the effective energy value of protein ingredients.

## 2. Materials and Methods

### 2.1. Diets and Ingredients

The basic diet (BD) was formulated to satisfy the nutritional needs of adult canines [21]. Table 1 shows the seven test diets, including the BD and diets involving 30% substitution of the BD, each replaced with fish meal (FM), meat and bone meal (MBM), corn gluten meal (CGM), soybean meal (SM), mealworm meal (MM), and yeast extract (YE), respectively. All diets were mixed uniformly in powder form.

### 2.2. Animals, Housing, and Experimental Design

The experiment took place at the companion animal testing center of the Special Animal and Plant Sciences of the Chinese Academy of Agriculture Sciences (Changchun, China). Beagles were kept in indoor enclosures covering floor space, adhering to prescribed light cycles, temperatures, and sanitation practices following the Animal Welfare Act guidelines. Before the experiment, all dogs had undergone deworming and vaccination, and no medications were administered throughout the study [20].

Throughout the experiment, except for the fasting period, dogs were provided with two meals of equal size at 09:00 and 14:00, with unrestricted access to fresh water. And the daily food intake of each beagle was recorded. All diets were provided as a mixture blended with water; the ratio between powder and water was 1:2.

The average weight of the six healthy adult female beagles was 15.07 ± 2.15 kg, and their body condition score (BCS) ranged from 4.5/9 to 5.5/9 [39]. The six dogs were each fed one of the seven diet treatments, according to a 7 × 6 incomplete Latin square design.

The beagles were individually housed in respiration chambers with a volume of 0.42 m^3^ [40]. Indirect calorimetry was performed as described by Zhang et al. [20] and conducted for seven periods. Each experiment period lasted for 10 days, including a 3-day adaptation period followed by a 7-day testing period (including a 4-day feeding period and a 3-day fasting period). Between each experiment period, have a 7-day washing period fed on BD. The beagles were weighed at the start of the feeding period and at the start and end of the fasting period. At 09:00 a.m. on d 0 of each experiment period, beagles were transferred to the chamber to adapt. Each dog was changed into a living chamber in each experiment period in proper order. Throughout the 7-day testing period, O_2_ consumption and CO_2_ production volumes were measured continuously for 4 consecutive days to assess total heat production (THP) and 3 consecutive days to assess fasting heat production (FHP), employing the Brouwer equation [41].

### 2.3. Fecal Score

During the feeding period, the fecal samples of each dog were scored every day. Fecal score was used using the following 5-point system: 1 = very hard, dry pellets. 2 = hard, formed, remains firm and soft; 3 = soft, formed, retains shape; 4 = unformed stool, pasty and slushy; and 5 = watery diarrhea [42,43]. The ideal fecal score was 2 to 3, indicating well-formed stools that were convenient to collect without being excessively dry [44].

### 2.4. Sample Collection

During the feeding period, total feces from each dog were weighed and collected once daily for 4 days. All fecal samples were stored at −20 °C. At the end of each experiment period, feces samples from each dog during each feeding period were uniformly mixed and dried at 65 °C, then smashed and sifted with a 1 mm screen before chemical analysis. 

Urine was collected daily during the 7-day testing period and then mixed with 10 mL of 10% sulfuric acid and measured for volume. Urine samples were separately mixed at the end of the feeding and fasting periods for each dog with each testing period, and then stored at −20 °C until analysis.

### 2.5. Chemical Analyses

Diets, ingredients, and feces were analyzed for dry matter (DM) (AOAC method 934.01 [45]). Nitrogen in all the diets, ingredients, feces, and urine samples was determined using the standard procedure (AOAC method 984.13 [45]), and crude protein (CP) was calculated as nitrogen × 6.25. The ether extract (EE), ash, calcium (Ca), phosphorus (P), and amino acids (AAs) contents in the diets, ingredients, and fecal samples were analyzed with AOAC 920.39, 967.05, 968.08, 985.01, and 994.12 [45]. The gross energy (GE) in the diets, ingredients, feces, and urine samples was determined using an adiabatic bomb calorimeter (IKA C2000, Staufer, Germany), with benzoic acid employed as the standard. The aflatoxin B1 and vomitoxin contents of the ingredients were determined by the use of ELISA kits (Sinobestbio Co., Ltd., Shanghai, China).

### 2.6. Calculations

The apparent total tract digestibility (ATTD) of energy and nutrients of test diets was calculated using the following equation:ATTD (%) = ((total intake of energy (kJ) or nutrients (g) − total fecal output of energy (MJ) or nutrients (g))/total intake of energy (kJ) or nutrients (g)) × 100%.

The ATTD and effective energy value of test ingredients were calculated as previously described by Adeola [46]:Ingredient digestibility (ID) % = (TDD − (1 − X) × BDD)/X,
Ingredient value (IE) MJ/kg DM = (TDE − (1 − X) × BDE)/X,
where ID, TDD, and BDD were the apparent digestibility of the ingredients, test diets, and BD, respectively (%); IE, TDE, and BDE were the energy value of the ingredients, test diets and BD, respectively, (MJ/kg DM); and X was the substitution ratio of the ingredients.

The values of DE, ME, and NE in the diet were calculated as follows [26]:DE = GE − fecal energy (FE),
ME = DE − urinary energy (UE),
NE = ME − heat increment (HI).

The THP and HI of beagles were determined using the following equations [41]:HI kJ/d = total heat production (THP) − fasting heat production (FHP),
THP or FHP kJ/d = 16.1753 VO_2_ (L) + 5.0208 VCO_2_ (L) − 5.9873 urinary N (g),
where VO_2_ was O_2_ consumption, and VCO_2_ was CO_2_ production. To account for the effect of body weight on energy metabolism and respiration between animals, the data were converted to metabolic weight [20].

### 2.7. Statistical Analyses

The data were presented in the format of the mean ± SEM and analyzed by using one-way ANOVA for energy value, nitrogen balance, O_2_ consumption, and CO_2_ production. Distinctions among diets or ingredients were assessed through Duncan’s multiple range test, with a significance level set at *p* < 0.05. Pearson’s correlation analysis was conducted to explore associations among various nutrients, energy values of ingredients, and equations. The estimation of equations was conducted using multiple linear regression through the stepwise method in SPSS 25.0 (SPSS Inc., Chicago, IL, USA). A graphical representation of correlation coefficients was generated using GraphPad Prism 9.0 software.

## 3. Results

### 3.1. Nutrient Composition of Test Ingredients 

The analyzed chemical composition of ingredients (DM basis) is shown in Table 2. The analyzed content of CP in the six test ingredients is listed in decreasing order as MM, FM, CGM, MBM, SBM, and YE, and all the test ingredients had a protein level greater than 40%. The concentrations of ash 34.83%, Ca 12.77%, and P 5.43% were found to be greater in MBM than in the other ingredients. Compared with CGM, SBM, and YE, MM, MBM, and FM had a greater EE content.

CGM had the highest gross energy content of 22.76 MJ/kg, while MBM had the lowest at 16.13 MJ/kg. Among the test ingredients, MM contained the highest levels of cysteine, threonine, arginine, valine, and leucine; FM was higher in lysine, histidine, and isoleucine; and CGM had the highest methionine, tyrosine, and phenylalanine content, which matched the higher CP content of the ingredients.

### 3.2. The Energy Values and the ATTD of GE and Nutrients of Diets

The ATTD of CP in the MBM diet was significantly lower than that of the BD, FM, and CGM diets (*p* < 0.05) (Table 3). The ATTD of CF in beagles fed FM and SBM diets was lower when compared with other diets (*p* < 0.05). The ATTD of DM in the MBM diet was significantly lower than in other diets (*p* < 0.05). Moreover, the ATTD of organic matter (OM) and GE in MM was the lowest among the diets. 

In terms of the energy value content of the test dietary diets, the gross energy of the MM diet was higher than that of other diets (*p* < 0.05). The FE values of the BD, FM, and CGM diets were significantly lower than those of other diets (*p* < 0.05). The UE of the CGM diet was the highest at 0.95 MJ/kg, significantly higher than the BD, FM, and MBM diets (*p* < 0.05).

The MM diet had the highest levels of DE and ME at 18.46 MJ/kg and 17.80 MJ/kg, while the MBM diet had the lowest at 13.31 MJ/kg and 12.61 MJ/kg. No significant variations were observed in NE between the BD, FM, and MM diets (*p* > 0.05).

The energy conversion efficiency of the ME:GE ratio of the FM diet was significantly greater than the MBM, SBM, and YE diets (*p* < 0.05). There were no significant differences seen for the ME:DE and NE:ME ratios (*p* > 0.05). The ME:DE ratio ranged from 94.63 to 97.48% among the seven diets, while the range of NE:ME is 75.47% to 86.07%. The ratios of NE:ME of the BD, FM, and MM diets were all above 80%.

### 3.3. Nitrogen Balance and Heat Production for Different Diets in Beagles

The data on the effects of test diets on nitrogen balance and heat production in beagles are presented in Table 4. No significant effect was observed for ME intake among the diets (*p* > 0.05). THP and HI were unaffected by the diets (*p* > 0.05). The HI of the diets listed in descending order as the YE, CGM, MBM, SBM, FM, and MM diets, and BD as the lowest one. There were no effects of NI, UN, RN, NPU, or PBV among the diets (*p* > 0.05). The FN of BD was significantly lower than the FM, MBM, CGM, MM, and YE diets (*p* < 0.05).

### 3.4. The Energy Values and the ATTD of Nutrients of the Test Ingredients

The ATTD of nutrients, as well as the DE, ME, and NE content of test ingredients, are shown in Table 5. Beagles fed FM, CGM, SBM, and YE had greater ATTD of DM and OM compared to those fed MM (*p* < 0.05). No distinctions were observed in the ATTD of CP and GE between the MBM and MM (*p* > 0.05), but they were lower compared to the other four ingredients (*p* < 0.05). The ATTD of CF in SBM was significantly lower than MBM, CGM, MM, and YE (*p* < 0.05). Overall, the ATTD of nutrients among the six ingredients was the lowest for MBM and MM and the highest for FM and CGM.

The energy value of the six ingredients was significantly different *(p* < 0.05). The DE values (MJ/kg DM) in descending order were MM at 22.95, CGM at 17.46, FM at 16.48, SBM at 15.36, YE at 15.11, and MBM at 6.73, and MM was significantly higher in comparison to the remaining five ingredients (*p* < 0.05). The ME content of MBM was significantly lower in comparison to the other five ingredients (*p* < 0.05). The NE of the FM, CGM, and MM were higher than that of the MBM (*p* < 0.05). In terms of energy utilization efficiency for the test ingredients, the ME:DE ratio ranged from 68.85% to 97.25%, with MM being significantly lower compared to the other ingredients (*p* < 0.05), and the NE:ME ratio ranged from 60.86% to 94.42%.

### 3.5. The Prediction Equations of the Energy Values of Poultry By-Product Meal, Fish Meal, Meat and Bone Meal, Corn Gluten Meal, Soybean Meal, Mealworm Meal, and Yeast Extract for Beagles

Combined with the findings of earlier research conducted by our team [20], the correlation between nutrient composition and DE, ME, and NE content of PBM and the six protein ingredients tested is presented in Figure 1. The ash content exhibited a negative correlation with OM (*p* < 0.01). A negative correlation (*p* < 0.01) was observed between the content of carbohydrate and CP (*p* < 0.01) and EE content. The content of CF exhibited a negative correlation with EE (*p* < 0.01) and a positive correlation with carbohydrates (*p* < 0.01).

The DE value demonstrated a positive correlation with ME, NE, and CP (*p* < 0.01) and a negative correlation with ash content (*p* < 0.01). The ME value showed a positive correlation with NE and OM (*p* < 0.01) and a negative correlation with ash (*p* < 0.01). The NE content was positively correlated with the CP content (*p* < 0.01).

Based on energy values and chemical composition, stepwise regression analysis was conducted to establish predictive equations for the effective energy, such as DE, ME, and NE (MJ/kg DM), of the seven ingredients, as shown in Table 6. The GE was the first predictor of DE content with R^2^ = 0.889 and RSD = 1.487 (*p* < 0.001), however, the precision of the equation was enhanced when CF was involved in the predictive equation with R^2^ = 0.964 and RSD = 0.845 (*p* < 0.001). The DE content had a strong correlation with the ME content, so it could be used as the only predictor in the ME prediction equations, where R^2^ = 0.799 and RSD = 0.117 (*p* < 0.001).

Protein and fiber content can serve as predictors of the effective energy value content of the ingredients. The prediction equations for DE, ME, and NE of the seven diets were: DE = 26.991 − 0.521ash − 0.143CHO − 0.446CF + 0.266EE where R^2^ = 0.964 and RSD = 0.845; ME = 16.521 − 0.267ash − 0.319GE − 0.287CF + 0.16CP where R^2^ = 0.919 and RSD = 0.899; and NE = 0.303 + 0.212CP − 0.154EE − 0.146CF where R^2^ = 0.930 and RSD = 0.582.

The deviations between the calculated values using the prediction equation in this study and the measured values of ME for MBM, MM, YE, CGM, FM, and SBM were all less than 10%, as shown in Table 7. When calculated using the NRC recommended equation, the deviations in ME for FM, CGM, SBM, MM, and YE compared to the measured were below 10%, with MBM reaching 67.13%. The differences between the calculated values using the prediction equation and the measured values of NE for six ingredients were all less than 10%.

### 3.6. Fecal Characteristics

Fecal characteristics, including fecal moisture content and fecal score, are shown in Table 8. Dogs fed the SBM diet had a higher fecal moisture content of 73.49% contrasted with the other diets (*p* < 0.05), and the MBM diet had the lowest fecal moisture content of 56.59% among the diets. The feces of the MBM diet contained more than 40% ash, and the other diets contained less than 20%. All fecal scores were within an acceptable range using the 5-point scale referenced previously. The YE diet had the highest fecal score of 3.19 compared with BD at 2.47 (*p* < 0.05) and did not differ among the FM, MBM, CGM, SBM, and MM diets (*p* > 0.05).

## 4. Discussion

### 4.1. Nutrient Composition of Test Ingredients and Diets

In the present study, we found that the digestibility and the energy value of plant protein ingredients and yeast extract were similar to those of animal protein, especially fish meal and poultry meal [20]. This means that recycled protein sources can normally be used in canine food. The reason for the difference between protein ingredients was analyzed mainly based on nutrition composition.

The chemical composition of the six ingredients fell within the scope of values reported in earlier reports [21,47,48,49,50,51]. Significant variations existed in the chemical composition of ingredients among those studies, with diversity, origin, and processing techniques identified as the principal contributors to these variations [52,53,54]. One example was the CP content in fish meal, which varied between 50% and 75% based on the variety and the origin of the fish [55,56,57]. The nutrient content of ingredients likewise changed depending on how they were processed, so the ether extract content of MM was reduced by 20% after degreasing [51], and the bioavailability of amino acids in MBM was affected after high temperature and high pressure [54].

Harmful mycotoxin in pet food ingredients could be a risk to pet health [58]. The levels of mycotoxins and vomiting toxins in all the ingredients were within the normal range specified in the *hygienic index and test method of pet feed* in China.

### 4.2. Energy Values and The ATTD of GE and Nutrients of the Test Ingredients

Nutrient digestibility acts as an indicator of the overall quality of the ingredients in canine diets [59]. The seven diets in this study not only provided adequate CP but also met the needs of adult medium-sized dogs [21,60]. Under these experimental conditions, beagles vary in their digestive utilization of different types of protein ingredients, as reported by Sieja et al. [61]. Animal by-product meals show considerable variability in digestibility. As one animal by-product used in pets, MBM contains abundant protein and energy [62]. The nutrient content is highly variable and easily dopant with bone meal, resulting in an excessive amount of ash and calcium, leading to an imbalanced calcium-phosphorus ratio and poor digestibility [63,64]. The lowest ATTD of DM in the MBM diet in this study was related to the elevated ash content of the ration, impacting the nutrient digestion in dogs [54], whereas the fecal energy value of 5.04 MJ/kg DM of MBM was the highest, aligning with the above results.

In general, plant-based ingredients exhibited a more consistent composition compared to animal-derived products but might be deficient in certain essential amino acids [65]. The ATTD of OM and GE of the CGM diet were consistent with those shown by Kawauchi et al. [66], and the digestibility of nutrients was higher than that of the SBM diet. The lack of sulfur-containing amino acids, such as methionine in pulses, and anti-nutritional factors affected digestibility and energy utilization. Chloe et al. also found low digestibility of pulse-based diets for beagles, whose protein digestibility ranged from 72% to 81% [67].

The energy values and utilization of the MM diet were opposite to the lower digestibility of OM and GE, which might be attributed to the increased chitin content in MM leading to decreased palatability [68]. The lower ME intake of 679.50 KJ/kg BW^0.75^/d in the MM diet of beagles also supported this observation. The higher energy utilization efficiency could be due to beagles adjusting the utilization form of effective energy in the diet to increase energy utilization efficiency, compensating for insufficient nutrient intake during feeding. Research by Deng et al. [69] reported that the energy metabolism rate of lambs in the 45% feeding *ad libitum* group was higher than that in the feeding *ad libitum* group, which is consistent with the result in this study. Protein quality is characterized by the capacity of dietary protein to meet the requirements for regular metabolism and maintenance of the body, and nitrogen metabolism, such as NI and retained nitrogen (RN), is mainly related to the level of CP and the quality of feed [70,71]. In this experiment, the BD diet contained less protein than the other diets, and NI and RN varied with the content of protein in diets. Dogs can adapt the protein concentration in diets that vary from 18 to 40%, and there was no difference in PBV and NPU among the test diets. These results were the same for mink [72], where the dietary protein content did not affect nitrogen utilization by animals within an appropriate range. As a new protein ingredient for pets, MM can be hydrolyzed to improve the palatability of foods [73,74,75]. The GE of the MM diet at 19.96 was higher than that of the BD diet at 18.32 MJ/kg, but the ME intake was less at 679.50 compared with 771.61 KJ/ kg BW^0.75^/d, probably due to differences in palatability resulting from differences in processing methods and in the proportion of additions, which affect the food intake of beagles.

Yeast has a role in regulating gastrointestinal function in dogs [15], where 2.00% YE increased cats’ food intake by 28.92% [76]. The ME intake of the YE diet in the present study increased by 5.60% compared to BD, which is consistent with the results of the above research. The HP and THP for beagles were not affected by the test diets, which was similar to the results of Li’s studies [77,78].

The nutritional content of different protein ingredients affects the utilization of energy. The low energy utilization observed in MBM evidenced the influence of the trail design on the outcomes, where the methodology used here with an oversupply of ash resulted in lower digestion of nutrients in the MBM diet provided the DE, ME, and NE were lower because the level of effective energy is a comprehensive reflection of the digestibility of chemical conventional nutrients [79]. The ME of FM, CGM, and SBM in this present study was comparable to the findings reported in the previous study [21]. Ma et al. used a 20% substitution ratio to assess the effective energy content of fish meal for growing pigs, which was similar to the results obtained in this study, indicating that monogastric animals such as dogs and pigs can better absorb and utilize fish meal as a protein ingredient [80]. The SBM and CGM are plant-based protein diets where the proportion of carbohydrate is higher, while YE is also high in carbohydrate as well as being rich in β-glucan and α-xylan [81]. Another study has shown no notable distinction in the nutrient digestibility of corn flour and rye ingredients with high starch content for beagles [82], which was consistent with the DE.

The ME and NE of SBM, CGM, and YE were not significantly different in this experiment. The efficiency of the utilization of ME for NE depends on its chemical composition [83]. The ratio of ME to DE in MM was lower than the other ingredients because it was determined in this study that increasing the proportion of fiber-rich chitin and its amino acid imbalance led to increased urinary nitrogen loss, increased urinary energy, reduced energy digestibility, and digestive metabolic rate. In previous studies, it was found that the nutrient composition of MM varied greatly and the effective energy value for different dogs was different [56,84], which may be related to the difference in dog breed and age and the nutrient composition of the raw material.

### 4.3. Equations for Predicting the Energy Values of Protein Feedstuffs for Beagles

Due to the labor-intensive nature of animal feeding studies, predictive equations are widely employed to compute energy concentrations in pet foods. The utilization of chemical composition to forecast the energy values of ingredients has proven to be an efficient method for evaluating the energy content of feed ingredients [85], using previously published results on poultry by-product meal in beagles [20]. The effective energy values for beagles were computed by using the difference method for the six protein ingredients. Subsequently, prediction equations for the energy values of the seven ingredients were established by regression analysis, utilizing the measured values of the ingredients along with their conventional chemical compositions. The formulation of prediction equations for ingredients relies on factors such as source, chemical attributes, nutrient digestibility, GE, sample quantity, and interactions among these variables [86,87]. The ME was estimated by employing calculations derived from measured concentrations of GE, moisture, protein, fat, ash, and fiber [23]. The prediction equations for metabolizable energy in this study obtained similar results when the ingredients were analyzed and compared to the equations reported in a prior study [24], which used GE, EE, CF, CP, and moisture as coefficients.

We also summarized the predictive energy equations for protein feedstuffs in dogs, which were more precise than the equations recommended by the NRC for feedstuffs with high ash content [20]. Compared with NRC, the predictive equation of ME is closer to the ME determined by the difference method in this study, especially for MBM. The probable reason is that the high content of ash decreased the digestibility of GE [63,64], but predictive equations recommended by the NRC overestimated energy digestibility. So, the result of ME predicted by the NRC equation was much higher than the measured ME in high-ash ingredients. And the predictive equation of ME in this study could be more suitable for protein feedstuffs with an ash content greater than 30%.

The NE of six protein ingredients was determined using the indirect calorimetry method, which needs specific instruments, so it is necessary to establish prediction equations for the net energy system. For companion animals, models to predict the NE density of food are rare, with only the Asaro model of dietary NE in commercial diets [24]. The prediction equation in number 5 of Table 6 established under this experimental condition could accurately predict the net energy values of the seven test ingredients with a high R^2^ of 0.930 and a low deviation of the residuals (RSD = 0.582) in the equations.

By correlating the different nutrient components and the energy values of feedstuffs, this study showed a robust positive correlation existed between the NE value and the protein content of feedstuffs for beagles. The correlation analysis between different nutrients and energy values of the ingredients for beagles indicated a strong positive correlation between the NE value and protein content, which resembles the findings of research by Zhang et al. regarding the strong correlation between the NE and protein of soybean meal for growing pigs [88].

This study also found a negative correlation between the net energy value content of the ingredients and their ash and carbohydrate content, where the CP content was the first predictor for predicting NE values of protein ingredients and CF was the predictor of all the effective energy values of the ingredients, indicating that the equation’s best predictor was dependent on the samples of the ingredient [28,89].

### 4.4. Fecal Characteristics for Beagles

The type and content of protein ingredients in canine food have an impact on fecal quality [90], which is a crucial indicator for assessing the quality of food and the overall health of the canine [35]. The CP content of BD of 26.10% and the fecal score of 2.47 in this study were similar to those reported by El-Wahab et al. of 22.00% and 2.39, respectively [83]. The crude protein content of the remaining test diets in the present study was all above 30%, and their fecal scores were higher than BD, which supports a positive correlation between canine fecal scores and dietary protein content [20,43].

Different types of protein ingredients in canine food can also have an impact on fecal characteristics. Research indicates that the dietary ash content of a diet is related to the frequency of defecation in canines [91]. The MBM diet of this study produced lower fecal moisture content, which might be due to its relatively substantial ash content in the meat and bone meal of 34.83%, which led to a shortened digestion time and decreased digestibility of the feed, increasing fecal dry matter content. Bednar showed that adding plant-based protein ingredients such as soybean meal to the diet caused an increase in the content of fecal moisture and output in canines [92,93,94]. Antinutritional factors and oligosaccharides in leguminous protein feedstuffs can affect fecal quality in dogs [95,96]. In the present study, the moisture content of fecal in the SBM diet was higher than the rest of the diets, which aligns with the outcomes of those studies. The fecal moisture in the current study was consistent with Berzelius’s results [97], who noted that feeding dogs high-carbohydrate diets can increase their fecal moisture content.

## 5. Conclusions

The protein ingredients FM, CGM, SBM, MM, and YE for beagles were effectively evaluated using the difference method with a 30% replacement ratio for their effective energy values. From the perspective of fecal quality, the 30% ratio did not affect the health of beagles, but further analyses are required for protein ingredients with higher ash content, such as MBM. The energy values of CGM were similar to FM, and those of SBM were similar to YE. Predictive energy equations for protein ingredients were derived from the study, which was more precise than the predictive equation of ME recommended by the NRC when the ash content of the ingredient was more than 30% DM.

## Figures and Tables

**Figure 1 animals-14-01599-f001:**
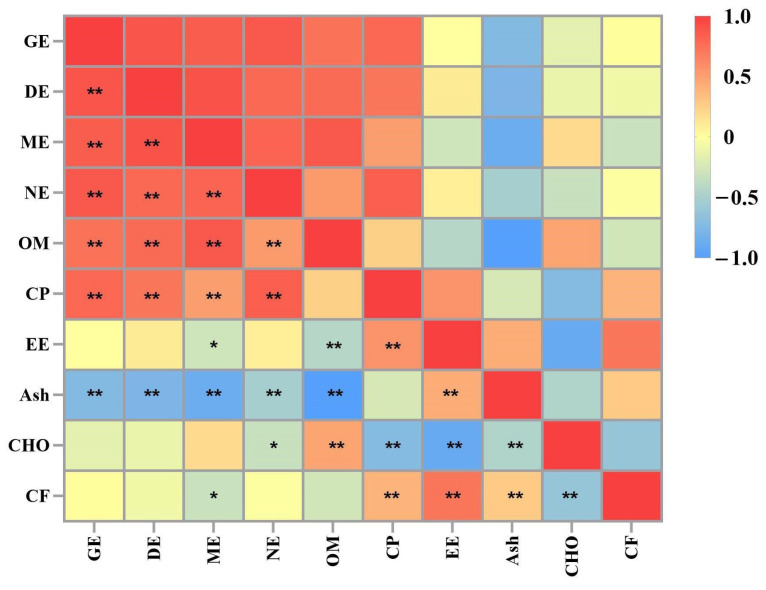
Correlation coefficients between chemical characteristics and energy utilization of seven ingredients in beagles. * *p* < 0.05; ** *p* < 0.01 (following Pearson’s correlation analysis). GE = gross energy; DE = digestible energy; ME = metabolizable energy; NE = net energy; OM = organic matter; CP = crude protein; EE = ether extract; CHO = carbohydrate; CF = crude fiber.

**Table 1 animals-14-01599-t001:** Ingredients and nutrient levels of the experimental diets. (% DM basis).

	Test Diets						
Item	BD	FM	MBM	CGM	SBM	MM	YE
Ingredients							
Broken rice	24.80	17.36	17.36	17.36	17.36	17.36	17.36
Sweet potato pellets	14.80	10.36	10.36	10.36	10.36	10.36	10.36
Potato starch	12.60	8.82	8.82	8.82	8.82	8.82	8.82
Soybean oil	10.00	7.00	7.00	7.00	7.00	7.00	7.00
Chicken meal	6.90	4.83	4.83	4.83	4.83	4.83	4.83
Fish meal	4.10	32.87	2.87	2.87	2.87	2.87	2.87
Beef and bone meal	3.20	2.24	32.24	2.24	2.24	2.24	2.24
Corn gluten meal	5.80	4.06	4.06	34.06	4.06	4.06	4.06
Soybean meal	6.70	4.69	4.69	4.69	34.69	4.69	4.69
Mealworm meal	0.00	0.00	0.00	0.00	0.00	30.00	0.00
Yeast extract	0.00	0.00	0.00	0.00	0.00	0.00	30.00
Chicken liver meal	3.15	2.21	2.21	2.21	2.21	2.21	2.21
Spray-dried blood cells	1.80	1.26	1.26	1.26	1.26	1.26	1.26
Beer yeast	0.90	0.63	0.63	0.63	0.63	0.63	0.63
Vitamin/mineral premix ^1^	3.00	2.10	2.10	2.10	2.10	2.10	2.10
Lysine	0.90	0.63	0.63	0.63	0.63	0.63	0.63
CaHPO4	0.81	0.57	0.57	0.57	0.57	0.57	0.57
Methionine	0.54	0.38	0.38	0.38	0.38	0.38	0.38
Total	100.00	100.00	100.00	100.00	100.00	100.00	100.00
Nutrient levels ^2^							
OM ^2^	92.42	86.98	84.90	93.90	92.74	91.84	92.77
GE (MJ/kg)	20.48	20.83	19.42	21.63	20.72	21.26	20.97
ME (MJ/kg) ^3^	13.84	13.95	13.00	14.42	14.00	14.09	14.35
CP	26.10	40.60	33.12	37.54	32.59	38.00	32.57
EE	11.41	11.82	11.48	8.19	9.29	11.00	8.91
Ash	7.58	13.02	15.10	6.10	7.26	8.16	7.23
Carbohydrate ^4^	54.91	34.56	40.31	48.18	50.86	42.84	51.29
CF	10.99	9.54	11.54	8.85	10.28	10.68	8.83

FM = fish meal; MBM = meat and bone meal; CGM = corn gluten meal; SBM = soybean meal; MM = mealworm meal; YE = yeast extract. OM = organic matter; GE = gross energy; ME = metabolizable energy; CP = crude protein; EE = ether extract; CF = crude fiber. ^1^ The premix provided the following per kg of diets: vitamin A 1,200,000 IU, Vitamin B1 1777.5 mg, vitamin B3 1672.5 mg, vitamin B6 1276.5 mg, vitamin B12 192.0 mg, vitamin D 546,768 IU, vitamin E 13,500 IU, *D*-biotin 34.5 mg, *D*-pantothenic acid 3205.5 mg, nicotinamide 18,034.5 mg, vitamin C 4500 mg, choline chloride 225,000 mg, Fe (as ferrous sulfate) 12,000 mg, Cu (as copper sulfate) 4050 mg, Mn (as manganese sulfate) 2100 mg, Zn (as zinc sulfate) 13,500 mg. ^2^ OM was calculated value, OM% = 100% − ash%. ^3^ ME was calculated value ME (kcal/100 g) = (5.7CP + 9.4EE 4.1CHO) × (91.2 – 1.43CF)/100 − 1.04CP. 1 kcal/100 g = 0.04184 kJ/kg. ^4^ Carbohydrate was calculated value. Carbohydrate% = 100% − CP% − EE% − ash%.

**Table 2 animals-14-01599-t002:** Chemical composition of fish meal, meat and bone meal, corn gluten meal, soybean meal, mealworm meal, and yeast extract. (%DM basis).

Item	FM	MBM	CGM	SBM	MM	YE
OM ^1^	80.42	65.17	89.28	93.48	91.20	92.90
GE (MJ/kg)	20.28	16.13	22.76	19.72	22.54	19.56
ME (MJ/kg) ^2^	14.26	10.93	14.71	13.86	15.83	14.98
CP	67.00	48.85	62.43	47.38	74.99	45.98
EE	8.19	12.34	0.79	1.29	15.41	0.58
Ash	19.58	34.83	10.72	6.52	8.80	7.10
Carbohydrate ^3^	5.23	3.99	26.07	44.81	0.80	46.34
CF	4.01	10.67	1.82	6.65	8.34	1.76
Ca	4.39	12.77	0.25	0.69	1.03	0.30
P	2.74	5.43	0.38	0.77	0.48	0.16
Aflatoxin B1 (μg/kg)	6.09	6.95	7.92	7.54	8.95	7.71
Vomitoxin (mg/kg)	1.70	1.82	1.41	1.69	1.61	1.55
Lysine	6.088	6.953	7.917	7.538	8.950	7.714
Methionine	1.697	1.816	1.411	1.693	1.613	1.549
Cysteine	5.785	2.785	1.330	3.371	4.993	3.176
Threonine	0.132	0.057	0.289	0.135	0.299	0.109
Tyrosine	0.142	0.100	0.356	0.245	1.131	0.246
Phenylalanine	2.786	1.352	2.037	1.901	2.961	1.861
Arginine	1.222	0.476	2.012	1.448	1.609	1.007
Histidine	3.195	1.805	4.323	2.900	3.438	2.031
Isoleucine	4.154	3.471	1.998	3.636	5.365	2.150
Leucine	2.064	0.986	1.634	1.485	1.129	1.031
Valine	3.365	1.432	2.874	2.655	3.237	2.176

FM = fish meal; MBM = meat and bone meal; CGM = corn gluten meal; SBM = soybean meal; MM = mealworm meal; YE = yeast extract. OM = organic matter; GE = gross energy; ME = metabolizable energy; CP = crude protein; EE = ether extract; CF = crude fiber. ^1^ OM was calculated value. OM% = 100% − ash%. ^2^ ME was calculated value. ME (kcal/100 g) = (5.7 CP + 9.4EE + 4.1Carbohydrate) × (91.2 − 1.43CF)/100 − 1.04CP. 1 kcal/100 g = 0.04184 kJ/kg. ^3^ Carbohydrate was calculated value. Carbohydrate% = 100% − CP% − EE% − ash%.

**Table 3 animals-14-01599-t003:** Effects of different diets on nutrient digestibility and energy value in beagles.

Item	BD	FM	MBM	CGM	SBM	MM	YE	SEM	*p*-Value
Digestibility coefficients (%)									
DM	79.49 ^b^	78.12 ^b^	68.98 ^d^	83.82 ^a^	77.66 ^b^	73.23 ^c^	77.38 ^b^	0.754	<0.001
OM	84.48 ^a^	84.35 ^a^	78.50 ^b^	83.54 ^a^	80.09 ^b^	75.38 ^c^	79.90 ^b^	0.590	<0.001
CP	75.52 ^a^	75.41 ^a^	68.99 ^c^	73.68 ^ab^	73.08 ^abc^	70.02 ^bc^	73.16 ^abc^	0.610	0.008
EE	95.29 ^a^	95.39 ^a^	93.44 ^b^	92.45 ^bc^	91.90 ^bc^	91.51 ^c^	91.13 ^c^	0.330	<0.001
GE	85.58 ^a^	85.51 ^a^	80.00 ^c^	83.99 ^a^	81.90 ^b^	77.04 ^d^	81.76 ^b^	0.493	<0.001
CF	77.23 ^a^	71.94 ^b^	79.16 ^a^	78.72 ^a^	72.07 ^b^	80.10 ^a^	76.60 ^a^	0.732	0.001
Energy values (MJ/kg DM)									
GE	19.86 ^b^	18.80 ^b^	18.35 ^b^	19.92 ^b^	20.06 ^b^	23.09 ^a^	19.72 ^b^	0.367	0.012
FE	2.54 ^b^	2.57 ^b^	5.04 ^a^	3.11 ^b^	4.34 ^a^	4.64 ^a^	4.46 ^a^	0.196	<0.001
UE	0.53 ^b^	0.58 ^b^	0.53 ^b^	0.95 ^a^	0.67 ^ab^	0.66 ^ab^	0.73 ^ab^	0.041	0.077
DE	16.13 ^b^	16.23 ^b^	13.31 ^c^	16.81 ^b^	15.72 ^b^	18.46 ^a^	15.26 ^b^	0.288	<0.001
ME	15.71 ^b^	15.65 ^b^	12.61 ^c^	15.85 ^b^	15.05 ^b^	17.80 ^a^	14.53 ^b^	0.325	<0.001
NE	13.50 ^ab^	12.84 ^abc^	10.12 ^c^	12.41 ^bc^	11.94 ^bc^	15.47 ^a^	10.97 ^bc^	0.405	0.006
Energy utilization (%)									
ME/GE	79.49 ^ab^	83.34 ^a^	68.42 ^c^	79.70 ^ab^	75.05 ^b^	77.41 ^ab^	73.88 ^bc^	1.011	0.001
ME/DE	97.48	96.55	94.68	94.63	95.79	96.43	95.20	1.168	0.996
NE/ME	85.82 ^a^	81.95 ^ab^	79.63 ^ab^	77.51 ^ab^	79.35 ^ab^	86.07 ^a^	75.47 ^b^	1.184	0.117

n = 6 adult dogs per treatment. FM = fish meal; MBM = meat and bone meal; CGM = corn gluten meal; SBM = soybean meal; MM = mealworm meal; YE = yeast extract. DM = dry matter; OM = organic matter; GE = gross energy; CP = crude protein; EE = ether extract; CF = crude fiber; FE = fecal energy; UE = urinary energy; DE = digestible energy; ME = metabolizable energy; NE = net energy. ^a–c^ In the same row: values with a different superscript are statistically different (*p* < 0.05).

**Table 4 animals-14-01599-t004:** Effects of different diets on nitrogen balance and heat production in beagles.

		Test Diets							
Item	BD	FM	MBM	CGM	SBM	MM	YE	SEM	*p*-Value
Energy balance (KJ/ kg BW^0.75^/d)									
ME intake	771.61	879.42	895.85	869.39	785.87	679.50	817.58	32.20	0.586
THP	665	691	708	730	674	675	738	21.29	0.964
HI	101 ^b^	125 ^ab^	148 ^ab^	164 ^ab^	131 ^ab^	113 ^ab^	182 ^a^	9.11	0.200
Nitrogen balance (g/ kg BW^0.75^/d)									
NI	1.69 ^b^	2.87 ^a^	2.82 ^a^	3.07 ^a^	2.48 ^ab^	2.63 ^a^	2.68 ^a^	0.12	0.057
FN	0.41 ^b^	0.70 ^a^	0.88 ^a^	0.81 ^a^	0.67 ^ab^	0.78 ^a^	0.71 ^a^	0.04	0.028
UN	0.88	1.10	1.27	1.39	1.18	1.18	1.14	0.07	0.678
RN	0.41 ^b^	1.08 ^a^	0.68 ^ab^	0.87 ^ab^	0.63 ^ab^	0.68 ^ab^	0.83 ^ab^	0.06	0.067
NPU (/%)	26.33	38.26	23.64	29.78	25.29	24.34	29.12	1.85	0.361
PBV (%)	34.55	50.55	34.42	40.50	34.72	34.60	39.94	2.49	0.553
Respiratory quotient									
Fed state	0.74	0.74	0.74	0.75	0.75	0.72	0.75	0.01	0.675
Fasted state	0.68	0.69	0.64	0.67	0.66	0.64	0.64	0.01	0.232

n = 6 adult dogs per treatment. FM = fish meal; MBM = meat and bone meal; CGM = corn gluten meal; SBM = soybean meal; MM = mealworm meal; YE = yeast extract; HI = heat increment; HP = heat production; THP = total heat production; NI = nitrogen intake; FN = fecal nitrogen; UN = urinary nitrogen; RN = retained nitrogen; NPU = net protein utilization; PBV = biological value of protein. ^a, b^ In the same row: values with a different superscript are statistically different (*p* < 0.05).

**Table 5 animals-14-01599-t005:** The nutrient digestibility and the energy values of fish meal, meat and bone meal, corn gluten meal, soybean meal, mealworm meal, and yeast extract in beagles.

	Ingredients							
Item	FM	MBM	CGM	SBM	MM	YE	SEM	*p*-Value
Digestibility coefficients (%)								
DM	74.93 ^a^	44.47 ^c^	79.86 ^a^	72.63 ^a^	57.88 ^b^	71.70 ^a^	2.46	<0.001
OM	71.70 ^ab^	69.89 ^b^	83.28 ^a^	71.77 ^ab^	56.08 ^c^	71.17 ^ab^	2.00	0.002
CP	75.15 ^a^	53.75 ^b^	72.62 ^a^	73.99 ^a^	62.56 ^ab^	73.06 ^a^	2.13	0.006
EE	94.38 ^a^	89.14 ^ab^	88.31 ^ab^	86.47 ^b^	85.17 ^b^	83.88 ^b^	1.04	0.044
GE	85.36 ^a^	65.58 ^c^	82.18 ^ab^	75.20 ^b^	59.01 ^c^	74.76 ^b^	1.84	<0.001
CF	59.58 ^bc^	78.50 ^a^	73.87 ^a^	51.70 ^c^	78.49 ^a^	66.82 ^ab^	2.26	<0.001
Energy values (MJ/kg DM)								
DE	16.48 ^b^	6.73 ^c^	17.46 ^b^	15.36 ^b^	22.95 ^a^	15.11 ^b^	1.03	<0.001
ME	15.71 ^a^	6.54 ^b^	15.80 ^a^	14.96 ^a^	15.83 ^a^	14.55 ^a^	0.57	<0.001
NE	12.54 ^a^	6.18 ^c^	11.21 ^ab^	9.00 ^bc^	11.13 ^ab^	8.90 ^bc^	0.51	0.029
Energy utilization (%)								
ME: DE	96.79 ^a^	97.25 ^a^	90.53 ^a^	97.71 ^a^	68.85 ^b^	96.69 ^a^	2.21	<0.001
NE: ME	79.96 ^ab^	94.42 ^a^	70.94 ^bc^	60.89 ^c^	70.37 ^bc^	60.86 ^c^	2.88	0.001

n = 6 adult dogs per treatment. FM = fish meal; MBM = meat and bone meal; CGM = corn gluten meal; SBM = soybean meal; MM = mealworm meal; YE = yeast extract. DM = dry matter; OM = organic matter; GE = gross energy; CP = crude protein; EE = ether extract; CF = crude fiber; FE = fecal energy; UE = urinary energy; DE = digestible energy; ME = metabolizable energy; NE = net energy. ^a–c^ In the same row: values with a different superscript are statistically different (*p* < 0.05).

**Table 6 animals-14-01599-t006:** The prediction equations of energy values (MJ/kg DM) of seven protein ingredients from chemical composition (% or MJ/kg DM) in beagles.

Num	Equations ^1^	R^2^	RSD	*p*-Value
1	DE = 4.727 + 0.446GE + 0.075CP + 0.225EE − 0.319ash	0.889	1.487	<0.001
2	DE = 26.991 − 0.521ash − 0.143CHO − 0.446CF + 0.266EE	0.964	0.845	<0.001
3	ME = 16.521 − 0.267ash − 0.319GE − 0.287CF + 0.16CP	0.919	0.899	<0.001
4	ME = 4.088 + 0.622DE	0.799	0.117	<0.001
5	NE = 0.303 + 0.212CP − 0.154EE − 0.146CF	0.930	0.582	<0.001
6	NE = −5.772 + 0.847GE	0.757	1.117	<0.001

OM = organic matter; GE = gross energy; ME = metabolizable energy; CP = crude protein; EE = ether extract; CF = crude fiber; DE = digestible energy; NE = net energy. ^1^ The value of the energy and chemical composition in the equations as dry matter basis. These prediction equations were established using 7 protein ingredients of beagles.

**Table 7 animals-14-01599-t007:** ME values and NE values of six protein ingredients in beagles were measured in this study and compared to those calculated using the NRC recommended equation and predictive equations. (MJ/kg DM).

Item	FM	MBM	CGM	SBM	MM	YE
Measured ME	15.71	6.54	15.80	14.96	15.83	14.55
Calculated ME ^1^	14.39	6.83	15.87	14.16	16.59	15.24
Delta ^2^	1.32	−0.29	−0.07	0.80	−0.76	−0.69
NRC Calculated ME ^3^	14.26	10.93	14.71	13.86	15.83	14.98
Delta ^2^	1.45	−4.39	1.09	1.10	0.00	−0.43
Measured NE	12.54	6.18	11.21	9.00	11.13	8.90
Calculated NE ^4^	11.49	6.51	12.12	8.60	11.43	8.81
Delta ^5^	1.05	−0.33	−0.91	0.40	−0.30	0.09

ME = metabolizable energy; FM = fish meal; MBM = meat and bone meal; CGM = corn gluten meal; SBM = soybean meal; MM = mealworm meal; YE = yeast extract; NE = net energy. ^1^ Predicted ME using the equation developed from the current experiment. The prediction equation is ME (MJ/kg) = 16.521 − 0.267ash − 0.319GE − 0.287CF + 0.16CP. ^2^ The difference between measured and estimated ME. ^3^ Predicted ME using NRC [20] equation. The NRC-recommended equation is ME (kcal/100 g) = (5.7CP + 9.4EE + 4.1 Carbohydrate) × (91.2 − 1.43CF)/100 − 1.04CP. 1 kcal/100 g = 0.04184 kJ/kg. ^4^ Predicted NE (MJ/kg) using the equation developed from the current experiment. The prediction equation is NE = 0.303 + 0.212CP − 0.154EE − 0.146CF. ^5^ The difference between measured and estimated NE.

**Table 8 animals-14-01599-t008:** The fecal quality of beagles with different diets.

	Test Diets								
Item	BD	FM	MBM	CGM	SBM	MM	YE	SEM	*p*-Value
Fecal moisture %	65.35 ^b^	58.61 ^bc^	56.59 ^c^	62.84 ^bc^	73.49 ^a^	62.85 ^bc^	63.33 ^bc^	1.12	<0.001
Fecal score	2.47 ^b^	2.78 ^ab^	2.97 ^a^	2.78 ^ab^	2.89 ^ab^	3.08 ^a^	3.19 ^a^	0.06	0.040

n = 6 adult dogs per treatment. FM = fish meal; MBM = meat and bone meal; CGM = corn gluten meal; SBM = soybean meal; MM = mealworm meal; YE = yeast extract. ^a–c^ In the same row, values with a different superscript differ significantly (*p* < 0.05). Fecal score: 1 = very hard, 2 = solid, well-formed “optimum”, 3 = soft, still formed, 4 = pasty, slushy, and 5 = watery diarrhea.

## Data Availability

Relevant data for this article can be obtained by contacting the author.

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
