# Peer review of "Evaluation of Effective Energy Values of Six Protein Ingredients Fed to Beagles and Predictive Energy Equations for Protein Feedstuff"

_animals, 2024, doi:10.3390/ani14111599_

Round 1
Reviewer 1 Report
Comments and Suggestions for Authors
Please check the paper for grammatical errors. There are few sentences which need to be rewritten.
Author Response
Response to Reviewers:
Reviewer 1
Comment 1: Line 20: repetition of Line 12, please change
Response: Thanks for your careful read and hard work on our previous submission. We have improved the sentences of Line 12. Details are shown as follows.
Line 12: The original sentence Protein ingredients have the largest proportion in pet food” has been changed to “Protein ingredients play a significant role in pet food and have been brought into focus by pet owners,” (Line 13)
Comment 2: Line 24: Suggested to change to “Predictive energy equations for protein ingredients were derived from the study”
Response: Thank you for pointing this out. By following your suggestions, we have changed the original sentence “Established the effective energy values prediction equations for protein ingredients.” into “The study resulted predictive energy equations for protein ingredients that were more accurate than the NRC's predictive equation of ME when the digestibility of energy was less than 70% and the crude fiber content exceeded 8% of the ingredients.” (Line 33-35)
Comment 3: Line 44. Check font of “ominivorous”
Response: Thanks for your careful read and hard work on our previous submission. We have corrected the careless font mistake in the manuscript.
Comment 4: Line 45: Complex sentence, please change
Response: Thanks for your suggestions. Thanks for your careful read and hard work on our previous submission. We have improved the sentence of Line 45. Details are shown as follows.
Line 45: The original sentence “One key aspect that pet food thought of the owner and the fabricators is the protein source and content, so protein ingredients have become an important component in pet foods.” has been changed to “One key aspect that pet food thought of the owners and the manufacturer is the protein source and content [3]. The content of protein in pet food is high, which in canine food ranges from 17.3% to 36.6% [4].” (Line 45 - 46)
Comment 5: Line 66-69: sentence is in different font, please check
Response: Thanks for your careful read and hard work on our previous submission. We have corrected the careless font mistake in the manuscript. (Line 70 - 72)
Comment 6: Line 77-80: repetitive sentences, please change
Response: Thanks for your suggestions. We have changed the sentence of Line 77-80. Details are shown as follows.
Line 77-80: The original sentence “Fish meal (FM), meat and bone meal (MBM), corn gluten meal (CGM), and soybean meal (SBM) are the traditional protein sources widely added to pet foods [20-22] and high-quality sustainable protein resources such as insect meals and yeast products have been used in pet foods in recent years [23-25].” has been changed to “Traditional animal protein ingredients, such as chicken, fish, and meat by-products, and plant proteins, such as soybean meal and corn gluten meal, have been the major protein ingredients used in the formulation of commercial pet foods [10-12]. In recent years, more sustainable high-protein novel ingredients, such as insect meal and mi-cro-algae, have also entered the pet food market [11,13].” (Line 61 - 64)
Comment 7: In the introduction section, please include some details about the TMM.
Response: Thanks for your suggestions. Thanks for your careful read and hard work on our previous submission. We have included some details about the mealworm (MM) of Line 73 - 74. Details are shown as follows.
Line 73 -74: “Mealworm meal is a kind of byproduct after Yellow mealworms (Tenebrio molitor) larvae defatted, which has a high quality and quantity of protein and amino acid profile. [16,17].”
Comment 8: Line 77-87: different font size, please check
Response: Thanks for your careful read and hard work on our previous submission. We have corrected the careless font mistake in the manuscript.
Comment 9: Table 1 can be moved to the results section since it shows the results of the chemical analyses.
Response: Thanks for your suggestions, we have changed the name of Table 1 and Table 2, and moved Table 2 to the result section.
The original Table 1 in Line 96 – 100 has been moved to Line 225 – 233.
Comment 10: For all the tables, please include the number of replicates as footnote
Response: Thanks for your suggestions, we have added the number of replicates as a footnote in Table 3, Table 4, Table 5, and Table 8.
The number of replicates “n=6 adult dogs per treatment.” has been added. (Line 255, Line 271, Line 287, Line 370)
Comment 11: Line 234-235: there is no line space between the footnote and next paragraph
Response: Thanks for your suggestions, we have added a line space between the footnote and the next paragraph.
Comment 12: Figure 1. Please include footnotes describing the acronyms used.
Response: Thanks for your suggestions, we have added the footnotes describing the acronyms used in Figure 1.
The acronyms used as footnotes “GE = gross energy; DE = digestible energy; ME = metabolizable energy; NE = net energy; OM = organic matter; CP = crude protein; EE = ether extract; CHO = Carbohydrate; CF = crude fiber.” have been added. (Line 313 - 315)
Comment 13: Line 436: check grammar
Response: Thanks for your careful read and hard work on our previous submission. We have deleted the careless mistakes. Details are shown as following.
Line 436: The original sentence “Based on our knowledge, this is the first time to reported the NE values of FM, MBM, CGM, SBM, TMM, and YE fed to beagles. The NE of six protein ingredients was determined using the indirect calorimetry (IC) method, which needs specific instruments, so it is necessary to establish prediction equations for the net energy system.” has been changed to “The NE of six protein ingredients was determined using the indirect calorimetry (IC) method, which needs specific instruments, so it is necessary to establish prediction equations for the net energy system.” (Line 493 - 495)
Comment 14: Line 446: please remove “that there was observed that”
Response: Thanks for your careful read and hard work on our previous submission. We have corrected the careless mistakes. Details are shown as following.
Line 453: The original sentence “By correlating the different nutrient components and the energy values of feedstuffs, this study showed that there was observed that a robust positive correlation existed between the NE value and the protein content of feedstuffs for beagles.” has been changed to “This study also found a negative correlation between the net energy value content of the ingredients and their ash and carbohydrate content, where the CP content was the first predictor for predicting NE values of protein ingredients and CF was the predictor of all the effective energy values of the ingredients indicating that the equation’s best predictor was depended on the samples of ingredient [28,90].” (Line 507 - 511)
Comment 15: Line 482: suggestion to change the sentence to “The ME: DE ratio ranged from 68.85 to 97.25% and the NE: ME ratio ranged from 60.86 to 94.42% respectively”.
Response: Thanks for your suggestions, we have rewritten the conclusion. Details are shown as following.
Conclusion
The protein ingredients FM, CGM, SBM, MM, and YE for beagles were effectively evaluated using the difference method with a 30% replacement ratio for their effective energy values. From the perspective of fecal quality, the 30% ratio did not affect the health of beagles, but further analyses are required for protein ingredients with higher ash content, such as MBM. The energy values of CGM were similar to FM and these of SBM were similar to YE. Predictive energy equations for protein ingredients were derived from the study which was more precise than the predictive equation of ME re-commended by NRC when the ash content of the ingredient was more than 30% DM. (Line 535 - 542)

Reviewer 2 Report
Comments and Suggestions for Authors
The study has evaluated nutrient value of different protein sources in adult Beagle dogs; chemical content, amino acid composition, digestibility, and energy measurements and N- balance, faecal score and equations for NE estimation based om chemical content.
The title does not cover what was done in the study. Much more was done than measure of NE values. Need to be rewritten.
Here are may comments: NE values are not considered in pet foods and pet food ingredients as it is not considered necessary as ME values are adequate. In production animals NE values are appropriate as feed cost is crucial. Nevertheless, NE values have scientific value for dogs also. So the energy evaluation of the protein ingredients in the study contributes with more knowledge in the field.
Line 12. Protein ingredients have the largest proportion in pet food. Is this true? ref...
Line 32. Not clear sentence: In conclusion..
Table 1. CF levels are higher than carbohydrates for some ingredients. Fish meal contains 3.66 % fiber, from where?
Table 5. CF digestibility was high . Comments?
You should consider to compare calculated ME values with Measured ME values in the discussion since you have both.
Comments on the Quality of English Language
The English language is good, but need another check from a native speaker.
Author Response
Response to Reviewers:
Comment 1: The title does not cover what was done in the study. Much more was done than measure of NE values. Need to be rewritten.
Response: Thanks for your careful comments, this question is very useful to us. The manuscript now has a new title, “Evaluation of effective energy values of six protein ingredients fed to beagles and predictive energy equations for protein feedstuff.”
Comment 2: Line 12. Protein ingredients have the largest proportion in pet food. Is this true? ref...
Response: Thanks for your careful read and hard work on our previous submission. We have corrected the careless mistakes. Details are shown as following.
Line 12: The original sentence “Protein ingredients have the largest proportion in pet food.” has been changed to “Protein ingredients play a significant role in pet food,” (Line 13)
Detailed relative reference are shown as following.
Swanson, K. S.; Carter, R. A.; Yount, T. P.; Aretz, J.; Buff, P. R. Nutritional sustainability of pet foods. Adv. Nutr. 2013, 4 (2), 141-150.
Comment 3: Line 32. Not clear sentence: In conclusion
Response: Thanks for your suggestions, we have corrected the sentence. Details are shown as follows.
Line 32-34: The original sentence “In conclusion, a correlation between the chemical composition and effective energy of protein ingredients, and recommended for use in determining the nutrient content of protein sources in dog food to predict its values.” has been changed to “In conclusion, the nutritional digestibility and the energy value of corn gluten meal were similar to fish meal and these of soybean meal were similar to yeast extract. All predictive energy equations for six protein feedstuffs had relatively slight differences with measured energy value.” (Line 36- 39)
Comment 4: Table 1. Fish meal contains 3.66 % fiber, from where?
Response: Thanks for your careful read and hard work on our previous submission.
The fish meal used in this experiment is made from whole fish, specifically redfish. It may contain undigested residue from the intestines which could affect the fiber content of the fish meal. Furthermore, Zheng used a fish meal similar to that used in this study in research on the digestive metabolism of common feedstuffs in adult Poodles, with a crude fiber content of 2.08%.
Detailed reference is presented as follows:
Zheng J. 2008. Study on nutrients digestibility and metabolism for common feedstuffs and protein requirement in adult Poodle dogs. Master. Gansu Agricultural University, Lanzhou, China, 2008.
Comment 5: Table 5. CF digestibility was high. Comments?
Response: Thanks for your careful review. Dainton et. al (2022) found that when the crude fiber content in the diet of adult female beagles reached 17.70%, it did not affect their nutrient digestibility or fecal quality. In this study, the highest dietary fiber content diet was 10.99%, indicating that beagles can effectively digest and absorb crude fiber within a certain range.
Detailed reference is presented as follows:
Dainton A N, He F, Bingham T W, et al. Nutritional and physico-chemical implications of avocado meal as a novel dietary fiber source in an extruded canine diet. J. Anim. Sci. 2022, ,100(2): skac026.
Comment 6: You should consider to compare calculated ME values with Measured ME values in the discussion since you have both.
Response: Thanks for your suggestions, this question is very useful to us. We have added a new Table (Table 7) in the result part. Compare measured ME values, and calculated ME values with the NRC recommended equation and prediction equation; compare measured NE values with calculated NE values based on the prediction equation in this study. And we have added some results and discussions about Table 7 in (Line 340 – 345), (Line 484 – 492).
Details are shown as following.
Table 7. ME values and NE values of six protein ingredients were measured in this study and compared to those calculated using the NRC recommended equation and predictive equations. (MJ/kg DM)
|
Item |
MBM |
MM |
YE |
CGM |
FM |
SBM |
|
Measured ME |
6.54 |
15.83 |
14.55 |
15.80 |
15.71 |
14.96 |
|
Calculated ME1 |
6.83 |
16.59 |
15.24 |
15.87 |
14.39 |
14.16 |
|
Delta2 |
-0.29 |
-0.76 |
-0.69 |
-0.07 |
1.32 |
0.80 |
|
NRC Calculated ME3 |
10.93 |
15.83 |
14.98 |
14.71 |
14.26 |
13.86 |
|
Delta2 |
-4.39 |
0 |
-0.43 |
1.09 |
1.45 |
1.1 |
|
Measured NE |
6.18 |
11.13 |
8.90 |
11.21 |
12.54 |
9.00 |
|
Calculated NE4 |
6.51 |
11.43 |
8.81 |
12.12 |
11.49 |
8.60 |
|
Delta5 |
-0.33 |
-0.30 |
0.09 |
-0.91 |
1.05 |
0.40 |
ME = metabolizable energy; FM = fish meal; MBM = meat and bone meal; CGM = corn gluten meal; SBM = soybean meal; MM = mealworm meal; YE = yeast extract; NE = net energy.
1 Predicted ME using the equation developed from the current experiment. The prediction equation is ME (MJ/kg) =16.521-0.267ash-0.319GE-0.287CF+0.16CP.
2 The difference between measured and estimated ME.
3 Predicted ME using NRC [20] equation. The NRC-recommended equation is ME (kcal/100 g) = (5.7CP + 9.4EE 4.1CHO) × (91.2 – 1.43CF)/100 – 1.04CP. 1 kcal/100 g = 0.04184 kJ/kg.
4 Predicted NE (MJ/kg) using the equation developed from the current experiment. The prediction equation is NE=0.303+0.212CP-0.154EE-0.146CF.
5 The difference between measured and estimated NE.
Line 340 – 345: The difference between the calculated values using the prediction equation and measured values of ME for MBM, MM, YE, CGM, FM, and SBM is less than 10%. When calculated using the NRC recommended equation, the difference in ME for FM, MBM, MM, and YE compared to the measured values exceeds 10%, with MBM reaching 91.74%. The difference between the calculated values using the prediction equation and the measured values of NE for six ingredients are all less than 10%.
Line 484 – 492: We also summarized the predictive energy equations for protein feedstuffs in dogs, which were more precise than the equations recommended by NRC for feedstuffs with high ash content [20]. Compared with NRC, the predictive equation of ME is closer to the ME determined by the difference method in this study, especially for MBM. The probable reason is that the high content of ash decreased the digestibil-ity of GE [63,64], but predictive equations recommended by NRC overestimated energy digestibility. So, the result of ME predicted by NRC equation was much higher than measured ME in high-ash ingredients. And the predictive equation of ME in this study could be more suitable for the protein feedstuffs with ash content more than 30%.

Reviewer 3 Report
Comments and Suggestions for Authors
I read your manuscript “Evaluation of effective energy values of six protein ingredients fed to beagles”. Dear Authors, I have carefully read your manuscript and in general, your article is poorly organized and requires a complete redesign. Currently it's too messy to be understood. Paper needs a lot of work. In my opinion, the manuscript needs careful review in writing before reaching journal standards. Some sample comments below:
Specific comments:
- The authors did not present any hypothesis at the beginning
- Background of the study should be made to clear.
- Provide more details of introduction and review of the work
Introduction
- to give data from the world, not only from China
- dogs are not as omnivores - are semi carnivores
- The equation for MER - recommended by NRC, not net energy (NE)
- add aim at the beginning
M&M
in table 1 ME was calculated - please complete the methodology by equations
in table 2 are Ca and P, for example in diet with fish meal Ca: P ratio is 3.09 : 1 , this is not the right ratio. according to the guidelines maximum 2: 1
add reference to Statistical analysis
Discussion: this is the weakest point of this article. A bit more elaboration
Conclusions: Need to be rewritten. An idea may be to synthetize in 3-5 bullet the key results of the study, evidences and recommendation. This improvement will increase clearness and readability. Add a practical implications statement
Author Response
Response to Reviewers:
Comment 1: The authors did not present any hypothesis at the beginning
Response: Thanks for your suggestions, this question is very useful to us.
The hypothesis of this study is: It is necessary to analyze different protein ingredients in pet food comprehensively and systematically to help pet food companies and pet owners know the utilization of environment-friendly protein sources. (Line 79 – 81)
Comment 2: Background of the study should be made to clear. Provide more details of introduction and review of the work
Response: Thanks for your suggestions, this question is very useful to us. We have added the background of the study and more details of the work in the introduction. Details are shown as following.
Introduction
The pet food production was 34.96 million tons all over the world in 2023, and it was still increasing while most of other animal feed was decreased [1]. Companion animals are always provided the best from their owners whose demand for high-quality pet food is very high [2]. One key aspect that pet food thought of the owners and the manufacturer is the protein source and content [3]. The content of pro-tein in pet food is high, which in canine food ranges from 17.3% to 36.6% [4]. To satisfy the standard of pet food as human food, some companies and some owners choose to not use or feed products containing byproduct meal [5]. Some pet foods do not contain plant protein and some pet owners have expressed concern that glutens in grains may be a source of allergies in dogs [6]. So, it is a big cost of animal protein originally sup-plied to humans in pet food. Attention to the environment, animal welfare, and climate change are encouraging institutions and individuals to seek alternatives to convention-al animal proteins.
Along with the 32,000-year history of the parallel evolution between dogs and humans and adapting an agricultural-based living conditions, dogs evolved from car-nivores to omnivores due to large changes in their food source [7]. Both animal ingre-dients and plant ingredients containing large amounts of protein and starch can be di-gested and absorbed by dogs [8,9]. To save animal protein-sourced protein and protect the environment, it would be a better way to use more plant-sourced protein and recy-cled protein to replace part of animal-sourced protein in pet food.
Traditional animal protein, such as chicken, fish, and meat by-products, and plant proteins, such as soybean meal and corn gluten meal, have been the major protein in-gredients used in the formulation of commercial pet foods [10-12]. In recent years, more novel sustainable ingredients with high protein, such as insect meal and sin-gle-cell protein, have also entered into the pet food market [11,13]. Yeast extract is the water-soluble extract produced from yeast waste streams, such as Saccharomyces cere-visiae, and separated from inner yeast cells. It could be a functional source of nutrients, since yeast extract is rich in proteins, amino acids, nucleotides, sugars, and a variety of trace elements [14]. Lin et al. showed that yeast products may be beneficial to adult dogs by positively altering gut microbiota, enhancing immune capacity, and reducing inflammation [15]. Because of insect proteins’ low land use, lower greenhouse gas emissions, and low water pollution, they may contribute to sustainable food production as an alternative source of animal protein. Mealworm meal is a kind of byproduct after yellow mealworms (Tenebrio molitor) larvae defatted, which has a high quality and quantity of protein and amino acids [16,17]. Pet food with insect-based ingredients was poorly accepted for human consumption to feed their pets. Insect-based pet foods proved to be attractive for purchase only when consumers are well informed about the product’s properties in terms of sustainability and healthiness for their pets [18]. So, it is necessary to analyze different protein ingredients in pet food comprehensively and systematically to help pet food companies and pet owners know the utilization of these environment-friendly protein sources.
Knowledge of the energy values and digestibility of ingredients is important to correctly balance pet food [19,20]. Current research on the effective energy value of pet food typically recommends the modified Atwater equation or predictive equations based primarily on fixed energy values and digestibility coefficients associated with the chemical composition of diets to estimate the metabolizable energy (ME) content of pet foods [20-23]. However, the equations do not apply to all ingredients and may overes-timate the food energy in animals [24,25]. The effective energy value of pet food is based on the ME energy system. In opposition to ME, net energy (NE) is a more precise evaluation of the true energy value of the feed [26], because it takes the heat increment (HI) from the digestion and metabolism of feeds into account [27,28]. We can find the rule of energy metabolism of different protein ingredients by total heat production (THP) and HI, it would be useful for losing weight pets and patients during nutrition recovery.
The most accurate method to assess the effective energy value of feed is to evalu-ate the animal's real digestive and metabolism condition in vivo. The difference method is more suitable for determining the nutrient digestibility and the effective energy val-ue of single ingredients in vivo [29-32]. A 30% substitution ratio in the difference method in vivo has been shown to effectively assess the energy content of poultry by-product meal for beagles in our previous study [20]. Traditional protein sources in pet foods include poultry by-product meal, fish meal (FM), meat and bone meal (MBM), corn gluten meal (CGM), and soybean meal (SBM) [32-34]. Recently, high-quality sus-tainable protein resources like insect meals and yeast products also have been used in pet foods [36-38].
This study aimed to determine the effective energy values of FM, MBM, CGM, SBM, mealworm meal (MM), and yeast extract (YE) by using the difference method, measure the nitrogen metabolism and heat production, and assess the feces score for beagles. By stepwise regression analysis of the measured energy value and chemical composition of ingredients fed to beagles, we also derived predictive equations for the effective energy value of protein ingredients.
Comment 3: to give data from the world, not only from China
Response: Thanks for your suggestions, this question is very useful to us. We have added data from the world. Details are shown as following.
The pet food production was 34.96 million tons all over the world in 2023, and it was still increasing while most of other animal feed were decreased [1]. (Line 43- 44)
Comment 4: dogs are not as omnivores - are semi carnivores
Response: Thanks for your careful read and hard work on our previous submission. We have corrected the careless mistakes. Details are shown as following.
Along with the 32,000-year history of the parallel evolution between dogs and humans and adapting an agricultural-based living conditions, dogs evolved from carnivores to omnivores due to large changes in their food source [7]. (Line 55- 57)
Comment 5: add aim at the beginning
Response: Thanks for your suggestions. We have added the aim of this study in Line 112 – 115. Details are shown as following.
This study aimed to determine the effective energy values of FM, MBM, CGM, SBM, mealworm meal (MM), and yeast extract (YE) by using the difference method, measure the nitrogen metabolism and heat production, and assess the feces score for beagles. By stepwise regression analysis of the measured energy value and chemical composition of ingredients fed to beagles, we also derived predictive equations for the effective energy value of protein ingredients. (Line 103- 108)
Comment 6: in table 1 ME was calculated - please complete the methodology by equations
Response: Thanks for your suggestions, this question is very useful to us. We have added the equation of ME in the footnote of Table 1.
The equation for ME is: ME (kcal/100 g) = (5.7CP + 9.4EE 4.1CHO) × (91.2 – 1.43CF)/100 – 1.04CP. 1 kcal/100 g =0.04184 kJ/kg. (Line 126- 127)
Comment 7: in table 2 are Ca and P, for example in diet with fish meal Ca: P ratio is 3.09 : 1 , this is not the right ratio. according to the guidelines maximum 2: 1
Response: Thank for your advice. The guidelines maximum 2: 1 comes from FEDIAF. The standard of base diet in the study followed NRC adult canine which did not limit the maximum of Ca: P ratio.
Comment 8: add reference to Statistical analysis
Response: Thanks for your suggestions, this question is very useful to us. Detailed reference is presented as follows:
Lyu, Z.; Chen, Y.; Wang, F.; Liu, L.; Zhang, S.; Lai, C., Net energy and its establishment of prediction equations for wheat bran in growing pigs. J. Anim. Biosci. 2023, 36 (1), 108.
Comment 9: Discussion this is the weakest point of this article. A bit more elaboration
Response: Thanks for your suggestions, this question is very useful to us. We have added some subheadings in the discussion part, making the discussion more readable.
Details are shown as following.
4.1. Nutrient Composition of Test Ingredients and Diets. (Line 377)
4.2. Energy Values and The ATTD of GE and Nutrients of The Test Ingredients. (Line 395)
4.3. Equations for Predicting the Energy Values of Protein Feedstuffs for Beagles. (Line 467)
4.4. Fecal Characteristics for Beagles. (Line 512)
Comment 10: Need to be rewritten. An idea may be to synthetize in 3-5 bullet the key results of the study, evidences and recommendation. This improvement will increase clearness and readability. Add a practical implications statement
Response: Thanks for your careful read and hard work on our previous submission. We have rewritten the conclusion part. Details are shown as following.
Conclusion
The protein ingredients FM, CGM, SBM, MM, and YE for beagles were effectively evaluated using the difference method with a 30% replacement ratio for their effective energy values. From the perspective of fecal quality, the 30% ratio did not affect the health of beagles, but further analyses are required for protein ingredients with higher ash content, such as MBM. The energy values of CGM were similar to FM and these of SBM were similar to YE. Predictive energy equations for protein ingredients were de-rived from the study which was more precise than the predictive equation of ME recommended by NRC when the ash content of the ingredient was more than 30% DM. (Line 535 - 542)

Round 2
Reviewer 3 Report
Comments and Suggestions for Authors
Dear Authors,
thank you for taking into account my comments in this new version of the paper which has certainly improved. The manuscript is now suitable for publication.